# α-Synuclein plasma membrane localization correlates with cellular phosphatidylinositol polyphosphate levels

**Reeba Susan Jacob[†], Cédric Eichmann[†], Alessandro Dema[‡], Davide Mercadante[§], Philipp Selenko***

Department of Biological Regulation, Weizmann Institute of Science, Rehovot, Israel

**Abstract** The Parkinson's disease protein α-synuclein (αSyn) promotes membrane fusion and fission by interacting with various negatively charged phospholipids. Despite postulated roles in endocytosis and exocytosis, plasma membrane (PM) interactions of αSyn are poorly understood. Here, we show that phosphatidylinositol 4,5-bisphosphate ($PIP_2$) and phosphatidylinositol 3,4,5-trisphosphate ($PIP_3$), two highly acidic components of inner PM leaflets, mediate PM localization of endogenous pools of αSyn in A2780, HeLa, SK-MEL-2, and differentiated and undifferentiated neuronal SH-SY5Y cells. We demonstrate that αSyn binds to reconstituted $PIP_2$ membranes in a helical conformation in vitro and that $PIP_2$ synthesizing kinases and hydrolyzing phosphatases reversibly redistribute αSyn in cells. We further delineate that αSyn-PM targeting follows phosphoinositide-3 kinase (PI3K)-dependent changes of cellular $PIP_2$ and $PIP_3$ levels, which collectively suggests that phosphatidylinositol polyphosphates contribute to αSyn's function(s) at the plasma membrane.

**\*For correspondence:**
philipp.selenko@weizmann.ac.il

[†]These authors contributed equally to this work

**Present address:** [‡]UCSF, School of Dentistry, Department of Cell and Tissue Biology, San Francisco, United States; [§]School of Chemical Sciences, University of Auckland, Auckland, New Zealand

**Competing interests:** The authors declare that no competing interests exist.

## Introduction

Aggregates of human α-synuclein (αSyn) constitute the main components of Lewy body inclusions in Parkinson's disease (PD) and other synucleinopathies (*Goedert et al., 2013*). In the brain, αSyn is abundantly found in different types of neurons, where it primarily localizes to presynaptic terminals and regulates synaptic vesicle (SV) clustering and trafficking (*Sulzer and Edwards, 2019*). Isolated αSyn is disordered in solution, whereas residues 1–100 adopt extended or kinked helical conformations upon binding to membranes containing negatively charged phospholipids (*Fusco et al., 2018*). Complementary electrostatic contacts between lysine residues within αSyn's N-terminal KTKEGV-repeats and acidic phospholipid headgroups align these α-helices on respective membrane surfaces (*Snead and Eliezer, 2019*). Membrane curvature (*Middleton and Rhoades, 2010*), lipid packing defects (*Nuscher et al., 2004*; *Pranke et al., 2011*) and fatty acid compositions (*Fortin et al., 2004*; *Galvagnion, 2017*) act as additional determinants for membrane binding. αSyn remodels target membranes (*Varkey et al., 2010*; *Westphal and Chandra, 2013*), which likely relates to its biological function(s) in vesicle docking, fusion and fission (*Sulzer and Edwards, 2019*). Furthermore, αSyn multimerization and aggregation may initiate at membrane surfaces, which holds important ramifications for possible cellular scenarios in PD (*Galvagnion, 2017*). Early αSyn oligomers bind to and disrupt cellular and reconstituted membranes (*Reynolds et al., 2011*; *Fusco et al., 2017*), whereas mature aggregates are found closely associated with membranous cell structures and intact organelles in cellular models of Lewy body inclusions (*Mahul-Mellier et al., 2020*) and in postmortem brain sections of PD patients (*Shahmoradian et al., 2019*).

Phosphatidylinositol phosphates (PIPs) are integral components of cell membranes and a universal class of acidic phospholipids with key functions in biology (*Balla, 2013*). Reversible phosphorylation of their inositol headgroups at positions 3, 4 and 5 generates seven types of PIPs, which act as

selective binding sites for folded and disordered PIP-interaction domains (*Balla, 2005*). In eukaryotic cells, PIPs make up less than 2% of total phospholipids with phosphatidylinositol 4,5-bisphosphate, PI(4,5)$P_2$ or PIP$_2$ hereafter, as the most common species (*McLaughlin et al., 2002*). PIPs function as core determinants of organelle identity (*Di Paolo and De Camilli, 2006*). PIP$_2$ is predominantly found at the inner leaflet of the plasma membrane (PM), where it acts as a signaling scaffold and protein-recruitment platform (*McLaughlin and Murray, 2005*). Carrying a negative net charge of $-4$ at pH 7 renders it more acidic than other cellular phospholipids such as phosphatidylserine (net charge $-1$) or phosphatidic acid (net charge $-1$) (*Kooijman et al., 2009*). Disordered PIP$_2$-binding domains contain stretches of polybasic residues that establish complementary electrostatic contacts with the negatively charged PIP head groups (*McLaughlin et al., 2002*) reminiscent of how αSyn KTKEGV-lysines interact with acidic phospholipids (*Dettmer, 2018*). Indeed, αSyn has been shown to bind to reconstituted PIP$_2$ vesicles in vitro (*Narayanan et al., 2005*). Phosphatidylinositol 3,4,5-trisphosphate, PI(3,4,5)$P_3$ or PIP$_3$ hereafter, harbors an additional phosphate group, which renders it even more acidic (net charge $-5$ at pH 7) (*Kooijman et al., 2009*). The steady-state abundance of PIP$_3$ at the PM is low (*Balla, 2013*), but local levels increase dynamically in response to cell signaling, especially following phosphatidylinositide-3 kinase (PI3K) activation (*Bilanges et al., 2019*).

Here, we set out to investigate whether native αSyn interacted with PM PIP$_2$ and PIP$_3$ in mammalian cells. Using confocal and total internal reflection fluorescence (TIRF) microscopy, we show that endogenous αSyn forms discrete foci at the PM of human A2780, HeLa, SK-MEL-2 and neuronal SH-SY5Y cells. The abundance and localization of these foci correlate with pools of PM PIP$_2$ and PIP$_3$. We further delineate high-resolution insights into αSyn interactions with reconstituted PIP$_2$ vesicles by nuclear magnetic resonance (NMR) spectroscopy and establish that αSyn binds PIP$_2$ membranes in its characteristic, helical conformation.

## Results

### PM localization of endogenous αSyn

To determine the intracellular localization of αSyn, we selected a panel of human cell lines (A2780, HeLa, SH-SY5Y and SK-MEL-2) that expressed low but detectable amounts of the endogenous protein. Confocal immunofluorescence (IF) localization in A2780 cells with an antibody that specifically recognizes αSyn without cross-reacting with its β and γ isoforms (*Figure 1—figure supplement 1A*) revealed a speckled intracellular distribution with distinct αSyn foci at apical and basal PM regions (*Figure 1A*). We verified overall antibody specificity by downregulating αSyn expression via siRNA-mediated gene silencing, which established that αSyn foci corresponded to endogenous protein pools (*Figure 1B* and *Figure 1—figure supplement 1B*, *Figure 1—figure supplement 1—source data 1*). To investigate colocalization of αSyn and PIP$_2$, we co-stained A2780 cells with αSyn and PIP$_2$ antibodies, and imaged basal PM planes by IF microscopy (*Figure 1C*, top panel). In 10–20% of cases, we detected clear superpositions of αSyn and PIP$_2$ signals, which we confirmed by measuring fluorescence intensity profiles over individual cell cross-sections (*Figure 1D*, top panel). We verified PM colocalization of αSyn with PIP$_2$ in SH-SY5Y cells that we differentiated into dopaminergic-like neurons following a stringent protocol and stimulation with retinoic acid (RA) and brain-derived neurotrophic factor (BDNF) (*Encinas et al., 2000*; *Figure 1—figure supplement 1C*, *Figure 1—figure supplement 1—source data 2*). We found prominent pools of αSyn in expanded structures reminiscent of synaptic boutons along neurites, where they colocalized with PIP$_2$ (*Figure 1C*, bottom panel, *Figure 1D*, and *Figure 1—figure supplement 1D*). These structures also stained positive for the presynaptic V-SNARE component syanptobrevin-2/VAMP2, a known binding partner of αSyn (*Burré et al., 2010*; *Figure 1—figure supplement 1E*).

To test whether changes in cellular PIP$_2$ levels affected αSyn abundance at the PM, we transiently overexpressed green fluorescent protein (GFP)-tagged phosphatidylinositol-4-phosphate 5-kinase PIPKIγ (*Krauss et al., 2006*). PIPKIγ localizes to the PM via a unique di-lysine motif in its activation loop (*Kunz et al., 2000*). Upon kinase expression, confirmed by GFP fluorescence, we detected increased amounts of αSyn at the PM of transfected cells (*Figure 1E*). By contrast, expression of GFP alone did not alter PM levels of αSyn. We obtained similar results in undifferentiated SH-SY5Y and HeLa cells (*Figure 1E*, *Figure 1—source data 1* and *Figure 1—figure supplement 2A*, *Figure 1—figure supplement 2—source data 1*) and confirmed that transient PIPKIγ expression

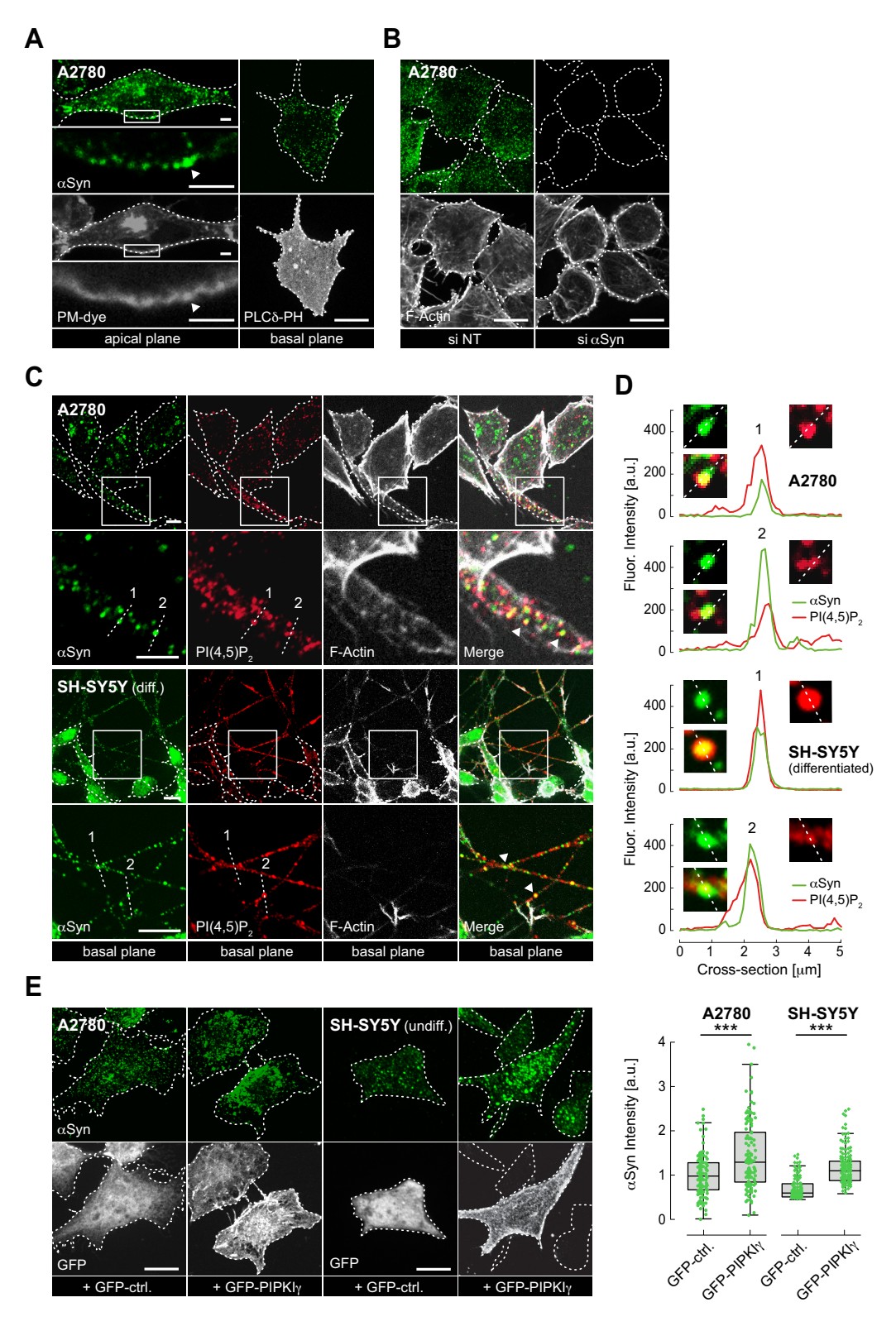

**Figure 1.** Plasma membrane (PM) localization of endogenous α-synuclein (αSyn). (**A**) Immunofluorescence detection of endogenous αSyn in A2780 cells by confocal microscopy. PM stained with tetramethylrhodamine-wheat germ agglutinin (WGA) (left panel) or identified via GFP-PLCδ-PH (right panel). Representative apical and basal confocal planes are shown. Scale bars are 2 µm (left) and 10 µm (right). (**B**) αSyn-PM localization in A2780 cells following control (si NT) and targeted siRNA (si αSyn) knockdown. Phalloidin staining of F-actin marks cell boundaries. Scale bars are 10 µm. (**C**)

*Figure 1 continued on next page*

*Figure 1 continued*

Immunofluorescence detection of endogenous αSyn and phosphatidylinositol 4,5-bisphosphate (PIP₂) at the PM in A2780 (top) and differentiated SH-SY5Y cells (bottom). Scale bars are 5 µm. (D) Spatially resolved αSyn (green) and PIP₂ (red) fluorescence intensity profiles across the dotted lines in the closeup views. Resolved αSyn and PIP₂ traces are marked with arrowheads. (E) αSyn-PM localization and quantification after transient green fluorescent protein (GFP) or GFP-PIPKIγ overexpression in A2780 and undifferentiated SH-SY5Y cells. GFP fluorescence identifies transfected cells. Scale bars are 10 µm. Box plots for αSyn immunofluorescence quantification. Data points represent n ~120 cells collected in four independent replicate experiments. Box dimensions represent the 25th and 75th percentiles, whiskers extend to the 5th and 95th percentiles. Data points beyond these values were considered outliers. Significance based on Student's *t* tests as ***p<0.001. See also *Figure 1—source data 1*.

The online version of this article includes the following source data and figure supplement(s) for figure 1:

**Source data 1.** Raw data of αSyn PM localization upon PIPKinase expression.

**Figure supplement 1.** αSyn siRNA knockdown, differentiation of SH-SY5Y cells and αSyn, PI(4,5)P2 and synaptobrevin 2/VAMP2 colocalization.

**Figure supplement 1—source data 1.** Uncropped western blots of αSyn siRNA knockdown.

**Figure supplement 1—source data 2.** Uncropped western blots of endogenous αSyn levels upon differentiation of SH-SY5Y cells.

**Figure supplement 2.** αSyn PM localization in A2780, HeLa, undifferentiated SH-SY5Y and SK-MEL-2 cells.

**Figure supplement 2—source data 1.** Quantification of αSyn PM levels upon GFP and GFP-PIPKIγ expression in HeLa cells.

**Figure supplement 2—source data 2.** Uncropped western blots of endogenous αSyn levels upon PIPKIγ expression.

**Figure supplement 2—source data 3.** Uncropped western blot of endogenous αSyn levels in A2780, HeLa, undifferentiated SH-SY5Y and SK-MEL-2 cells.

did not affect overall αSyn abundance (*Figure 1—figure supplement 2B*, *Figure 1—figure supplement 2—source data 2*). These findings suggested that PM localization of endogenous αSyn correlated with cellular PIP₂ levels. To better resolve the presence of αSyn at the PM, we resorted to TIRF microscopy. Employing a narrow evanescent field depth of ~50 nm, we detected endogenous αSyn at PM foci in A2780, HeLa, SH-SY5Y and SK-MEL-2 cells, which correlated with total αSyn levels determined by semi-quantitative western blotting (*Figure 1—figure supplement 2C*, *Figure 1—figure supplement 2—source data 3*). Importantly, both imaging approaches were targeted toward detecting PM pools of αSyn and did not aim at interrogating cytoplasmic fractions of the endogenous protein.

## αSyn binds reconstituted PIP₂ vesicles

To test whether αSyn directly bound PIP₂ membranes under physiological salt and pH conditions (150 mM, pH 7.0), we added N-terminally acetylated, $^{15}$N isotope-labeled αSyn to reconstituted PIP₂ vesicles. Circular dichroism (CD) spectroscopy revealed characteristic helical signatures (*Davidson et al., 1998*; *Jo et al., 2000*; *Figure 2A*), whereas NMR experiments confirmed site-selective line broadening of N-terminal residues 1–100, confirming membrane binding (*Bodner et al., 2009*; *Dikiy and Eliezer, 2014*; *Figure 2B* and *Figure 2—figure supplement 1*, *Figure 2—figure supplement 1—source data 1*). In line with these observations, we detected remodeled PIP₂ vesicles by negative-stain transmission electron microscopy (EM), manifested by tubular extrusions emanating from reconstituted specimens and agreeing with published findings on other membrane systems (*Varkey et al., 2010*; *Westphal and Chandra, 2013*; *Figure 2A*). Together, these results established that residues 1–100 of αSyn interacted with PIP₂ vesicles in helical conformations that imposed membrane remodeling, whereas its 40 C-terminal residues did not engage in membrane binding and remained flexible and disordered. To gain further insights into αSyn-PIP₂ interactions, we reconstituted phosphatidylcholine (PC)-PIP₂ vesicles (100 nm diameter) at a fixed molar ratio of 13:1 (PC:PIP₂) (*Figure 2C*). We added increasing amounts of these PC-PIP₂ vesicles to αSyn and measured CD and dynamic light scattering (DLS) spectra of the resulting mixtures. Up to a ~50-fold molar excess of lipid to protein, αSyn interacted with PC-PIP₂ vesicles in a helical conformation without disrupting the monodisperse nature of the specimens, that is, without membrane remodeling (*Figure 2C* and *Figure 2—figure supplement 2A*). In parallel, we performed NMR experiments on these samples and measured intensity changes of αSyn resonances in a residue-resolved manner (*Figure 2D* and *Figure 2—figure supplement 2B*, *Figure 2—figure supplement 2—source data 1*). Analyzing signal intensity ratios (I/I₀) of unbound (I₀) and PC-PIP₂-bound αSyn (I), we found that residues 1–10 constituted the primary interaction sites, whereas residues 11–100 displayed progressively weaker membrane contacts. In agreement with our experiments on PIP₂-only vesicles, we detected no contributions by C-terminal αSyn residues. These findings confirmed the

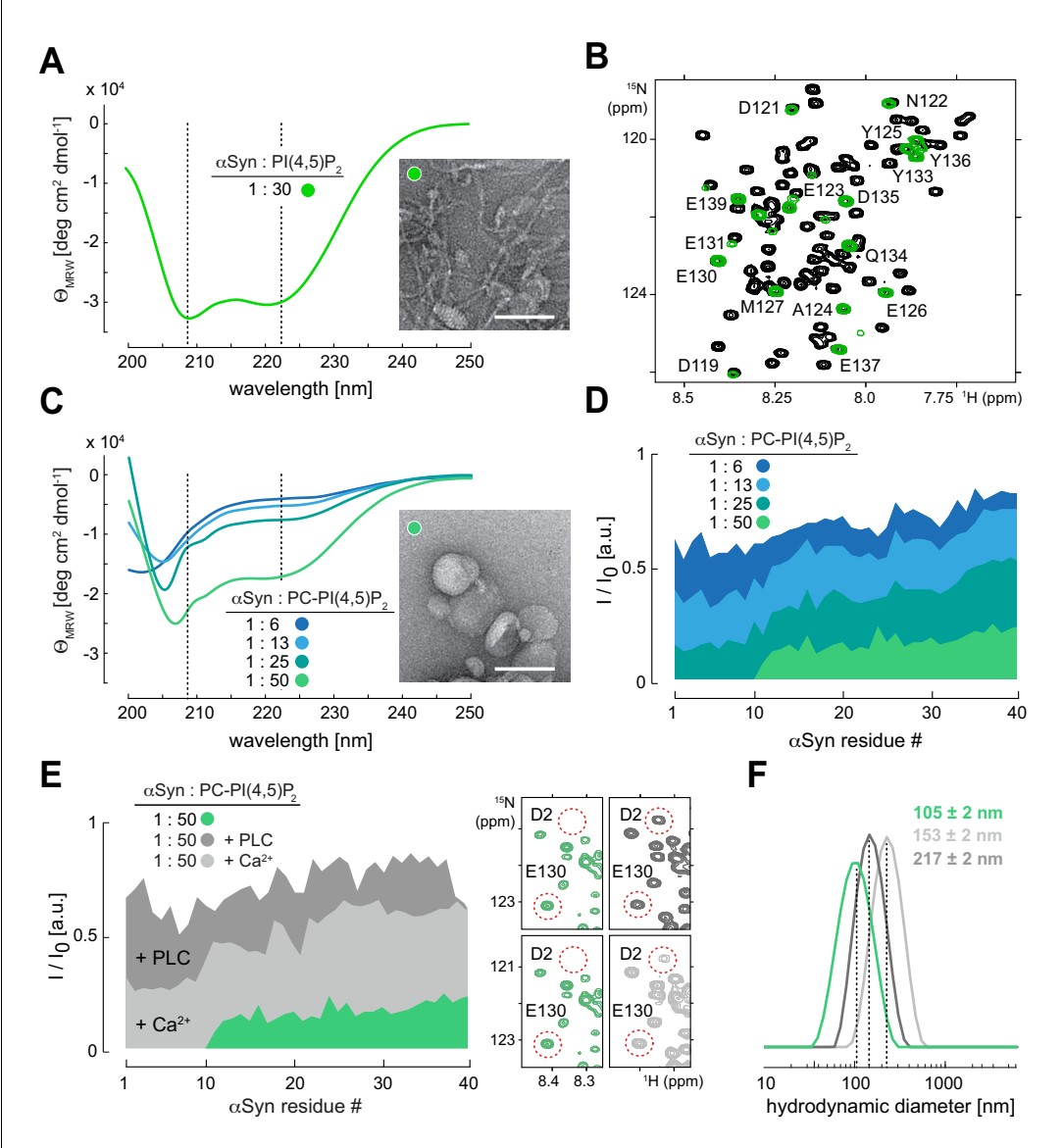

**Figure 2.** α-Synuclein (αSyn) binding to reconstituted phosphatidylinositol 4,5-bisphosphate (PIP$_2$) vesicles. (**A**) Circular dichroism (CD) spectrum and negative-stain electron micrograph of αSyn-bound PIP$_2$ vesicles (100%). Scale bar is 100 nm. (**B**) Overlay of 2D $^1$H-$^{15}$N nuclear magnetic resonance (NMR) spectra of isolated αSyn in buffer (black) and bound to PIP$_2$ vesicles (green). Remaining signals of C-terminal αSyn residues are labeled. (**C**) CD spectra of αSyn bound to phosphatidylcholine (PC)-PIP$_2$ vesicles at increasing lipid-to-protein ratios (inset) and negative-stain electron micrograph of the αSyn:PC-PIP$_2$ (1:50 protein:PIP$_2$) sample. Scale bar is 100 nm. (**D**) NMR signal intensity ratios of bound (I) over unbound (I$_0$) αSyn in the presence of different amounts of PC-PIP$_2$ vesicles (equivalent to (**C**)). Only residues 1–40 are shown. (**E**) I/I$_0$ of PC-PIP$_2$ bound αSyn at 1:50 (green) and after addition of phospholipase C (PLC) (dark gray) and Ca$^{2+}$ (light gray). Selected regions of 2D $^1$H-$^{15}$N NMR spectra of PC-PIP$_2$ bound αSyn (left, green) and in the presence of PLC (top right, dark gray) and Ca$^{2+}$ (bottom right, light gray). Release of N-terminal αSyn residues from vesicles and reappearance of corresponding NMR signals are indicated for Asp2 (D2) as an example. (**F**) Hydrodynamic diameters of αSyn-bound PC-PIP$_2$ vesicles before (green) and after PLC (dark gray) and Ca$^{2+}$ (light gray) addition by dynamic light scattering. Errors were calculated based on measurements on three independent replicate samples.

The online version of this article includes the following source data and figure supplement(s) for figure 2:

**Figure supplement 1.** NMR characterization of αSyn binding to reconstituted PI(4,5)P2 vesicles.

**Figure supplement 1—source data 1.** NMR signal intensity ratios of αSyn binding to PI(4,5)P2 vesicles.

**Figure supplement 2.** NMR characterization of αSyn binding to reconstituted PC-PI(4,5)P2 vesicles.

**Figure supplement 2—source data 1.** NMR signal intensity ratios of αSyn binding to PC-PI(4,5)P2 vesicles.

**Figure supplement 3.** NMR characterization of mutant αSyn binding to reconstituted PC-PI(4,5)P2 vesicles and upon phospholipase C (PLC) treatment.

**Figure supplement 3—source data 1.** NMR signal intensity ratios of αSyn binding to PC-PI(4,5)P2 vesicles upon phospholipase C (PLC) treatment.

*Figure 2 continued on next page*

*Figure 2 continued*

**Figure supplement 4.** NMR characterization of αSyn binding to reconstituted PC-PI(4,5)P2 vesicles in the presence of Ca and inositol polyphosphate (IP6) interaction.

**Figure supplement 4—source data 1.** NMR signal intensity ratios of αSyn binding to PC-PI(4,5)P2 vesicles upon Ca addition.

tri-segmental nature of αSyn-PIP$_2$ interactions and the importance of anchoring contacts by N-terminal αSyn residues, similar to other membrane systems (*Bodner et al., 2009*; *Fusco et al., 2014*; *Fusco et al., 2016*). To further validate our conclusions, we performed NMR experiments with mutant forms of αSyn in which we deleted residues 1–4 (ΔN) (*Bartels et al., 2010*), substituted Phe4 and Tyr39 with alanine (F4A-Y39A) (*Lokappa et al., 2014*), or oxidized αSyn Met1, Met5, Met116 and Met123 to methionine-sulfoxides (MetOx) (*Maltsev et al., 2013*; *Figure 2—figure supplement 3A*). In line with earlier reports, we did not observe binding to PC-PIP$_2$ vesicles for any of these variants. Our results corroborated that PC-PIP$_2$ interactions strongly depended on intact N-terminal αSyn residues, with critical contributions by Phe4 and Tyr39, and requiring Met1 and Met5 in their reduced states.

In contrast to other lipids, PIPs offer attractive means to regulate the reversibility of αSyn-membrane interactions. Different charge states of PIPs can be generated from phosphatidylinositol (PI) precursors by action of PIP kinases and phosphatases (*Matteis and Godi, 2004*), or via PIP conversion by lipases such as phospholipase C (PLC) to produce soluble inositol 1,4,5-trisphosphate (IP$_3$) and diacylglycerol (*Berridge and Irvine, 1984*; *Figure 2—figure supplement 3B*). To investigate the reversibility of αSyn-PIP$_2$ interactions, we prepared PC-PIP$_2$ vesicles bound to $^{15}$N isotope-labeled αSyn to which we added catalytic amounts of unlabeled PLC. We reasoned that PLC will progressively hydrolyze PIP$_2$ binding sites and, concomitantly, release αSyn. In turn, we expected to observe an increase of αSyn NMR signals corresponding to the fraction of accumulating, unbound protein molecules. Indeed, we detected the recovery of αSyn NMR signals upon PLC addition (*Figure 2E* and *Figure 2—figure supplement 3C*, *Figure 2—figure supplement 3—source data 1*). Next, we asked whether αSyn binding to PC-PIP$_2$ vesicles was sensitive to calcium, a competitive inhibitor of many protein-PIP$_2$ interactions (*Bilkova et al., 2017*). Whereas overall binding was greatly reduced, we found that the first 10 residues of αSyn displayed residual anchoring contacts with PC-PIP$_2$ vesicles even at high (2.5 mM) calcium concentrations (*Figure 2E* and *Figure 2—figure supplement 4A*, *Figure 2—figure supplement 4—source data 1*), confirming earlier results on the stability of αSyn PC-PIP$_2$ vesicle interactions in the presence of calcium (*Narayanan et al., 2005*). Notably, DLS measurements showed that hydrodynamic diameters of PC-PIP$_2$ vesicles expanded upon PLC treatment and in the presence of calcium, irrespective of whether αSyn was bound (*Figure 2F* and *Figure 2—figure supplement 4B*). This further suggested that vesicle remodeling and concomitant curvature reductions did not abolish αSyn interactions. Finally, we sought to determine whether electrostatic interactions with acidic PIP headgroups alone mediated αSyn binding. To this end, we added a fourfold molar excess of free inositol polyphosphate (IP$_6$) to $^{15}$N isotope-labeled αSyn. Surprisingly, we did not detect binding of αSyn to this highly negatively charged entity (*Figure 2—figure supplement 4C*), which insinuated that αSyn interactions with PIP-containing membranes required additional lipid contributions besides headgroup contacts.

## αSyn-PM localization correlates with changes in PIP$_2$-PIP$_3$ levels

Following these results, we asked whether reversible αSyn-PIP$_2$ interactions were present in cells. To answer this question, we transiently overexpressed different PM-targeted PIP phosphatases in A2780 cells and quantified PM localization of endogenous αSyn by confocal IF microscopy (*Figure 3A*, *Figure 3—source data 1*). Specifically, we expressed MTM1-mCherry-CAAX, which hydrolyzes PI(3)P to yield PI, INPP5E-mCherry-CAAX to produce PI(4)P from PIP$_2$ and PTEN-mCherry-CAAX to create PIP$_2$ from PI(3,4,5)P$_3$, as described (*Posor et al., 2013*). In agreement with our hypothesis, only the conversion of PIP$_2$ to PI(4)P by INPP5E led to a marked reduction of endogenous αSyn at the PM (*Figure 3A*). Together with earlier kinase results, these findings corroborated that PM localization of cellular αSyn was modulated by PIP$_2$-specific enzymes. Next, we asked whether signaling-dependent activation of PI3K and concomitant accumulations of the even more negatively charged PIP$_3$ (*Bilanges et al., 2019*) led to dynamic changes of αSyn abundance at the

PM. To this end, we employed histamine stimulation of SK-MEL-2 cells that we transiently co-transfected with histamine 1 receptor 1 (H1R) and a PH-domain GFP-fusion construct of the general receptor of phosphoinositides 1 (GRP1) that specifically interacts with membrane PIP$_3$ (*Kavran et al., 1998*). Because histamine-mediated PI3K activation also induces time-dependent secondary effects including PIP$_2$ hydrolysis by PLC (*Saheki et al., 2016*), we monitored αSyn localization and PIP$_2$-PIP$_3$ levels in a time-resolved fashion by fixing SK-MEL-2 cells at 40, 85, 120, and 240 s after histamine addition (*Figure 3B*, *Figure 3—source data 2*). After 40 s, we observed an initial increase of PIP$_2$ and PIP$_3$ levels at the PM, which was mirrored by greater pools of endogenous αSyn at basal membrane regions. While PIP$_2$ levels dropped at intermediate time points (85–120 s), likely due to PLC-mediated PIP$_2$ hydrolysis, PIP$_3$ concentrations were highest at 85 s and leveled off more slowly (120–240 s). Interestingly, PM-αSyn followed the observed PIP$_3$ behavior in a remarkably similar manner. At later time points (240 s), we noted a significant redistribution of cellular PIP$_2$ and PIP$_3$ pools toward the edges of SK-MEL-2 cells, coinciding with the accumulation of bundled actin fibers and in line with expected PI3K-signaling-dependent rearrangements of the cytoskeleton (*Bilanges et al., 2019*). αSyn colocalization with these peripheral PIP$_2$-PIP$_3$ speckles was significantly higher than at earlier time points (*Figure 3B* and *Figure 3—figure supplement 1A*). We independently confirmed these results with single time point measurements by TIRF microscopy (*Figure 3—figure supplement 1B*, *Figure 3—figure supplement 1—source data 1*). To investigate whether other PI3K pathways caused similar effects, we stimulated SK-MEL-2 with insulin, which triggers PI3K activation via receptor tyrosine kinase signaling (*Ruderman et al., 1990*). We verified that SK-MEL-2 cells endogenously expressed the insulin-like growth factor-1 receptor β (IGF-1 Rβ) by western blotting (*Figure 3—figure supplement 1C*, *Figure 3—figure supplement 1—source data 2*). In support of our hypothesis, we measured increased αSyn-PM localization by TIRF microscopy upon insulin stimulation for 10 min (*Figure 3—figure supplement 1D*, *Figure 3—figure supplement 1—source data 3*). Given the short exposure times to histamine and insulin in these experiments, we reasoned that observed PM accumulations likely reflected enhanced recruitment of existing αSyn pools rather than de novo protein synthesis and PM targeting, thus providing further evidence that αSyn abundance at the PM correlated with signaling-dependent changes of PIP$_2$ and PIP$_3$ levels.

## Discussion

Our results establish that clusters of endogenous αSyn are found at the PM of human A2780, HeLa, SK-MEL-2 and SH-SY5Y cells, where their abundance correlates with PIP$_2$ levels (*Figure 1*). Specifically, we show that targeted overexpression of the PIP$_2$-generating kinase PIPKIγ increases αSyn at the PM (*Figure 1E*), whereas the PIP$_2$-specific phosphatase INPP5E reduces the amount of PM αSyn (*Figure 3A*). We further demonstrate that PIP$_3$-dependent histamine and insulin signaling redistributes αSyn to the PM (*Figure 3B* and *Figure 3—figure supplement 1*), which collectively suggests that changes in PM PIP$_2$ and PIP$_3$ levels affect intracellular αSyn localization in a dynamic and reversible manner. Aiming for a stringent analysis, we investigated αSyn-PM interactions at strictly native, endogenous protein levels and intentionally refrained from transient or stable overexpression to not confound our analysis with non-physiological off-target effects. Moreover, we chose to study αSyn in an unaltered sequence context, that is, without modifying the protein with fluorescent dyes or fusion moieties. These requirements precluded live-cell imaging experiments to determine PM-localization kinetics, although histamine and insulin stimulation experiments suggest that endogenous αSyn pools redistribute readily. While we cannot rule out that additional secondary protein–protein interactions contribute to PM targeting, we demonstrate that αSyn directly interacts with reconstituted PIP$_2$ vesicles in vitro (*Figure 2*). Importantly, the biophysical characteristics of these interactions are indistinguishable from other previously identified, negatively charged membrane systems with primary contacts by N-terminal αSyn residues 1–10 and progressively weaker interactions along residues 11–100. The last 40, C-terminal residues of αSyn are not involved in PIP$_2$ membrane binding, similar to all other reconstituted vesicular or planar lipid surface interactions studied thus far (*Bodner et al., 2009*; *Fusco et al., 2014*; *Perrin et al., 2000*). Based on the known preferences for negatively charged phospholipids, PIP$_2$ and PIP$_3$ constitute intuitive αSyn binding partners. Not only because of their highly acidic nature (*Kooijman et al., 2009*), but also because of their acyl chain compositions containing saturated stearic-(18:0) and polyunsaturated arachidonic acids (20:4), the latter conferring 'shallow' lipid packing defects (*Bigay and Antonny, 2012*) that are ideally suited to

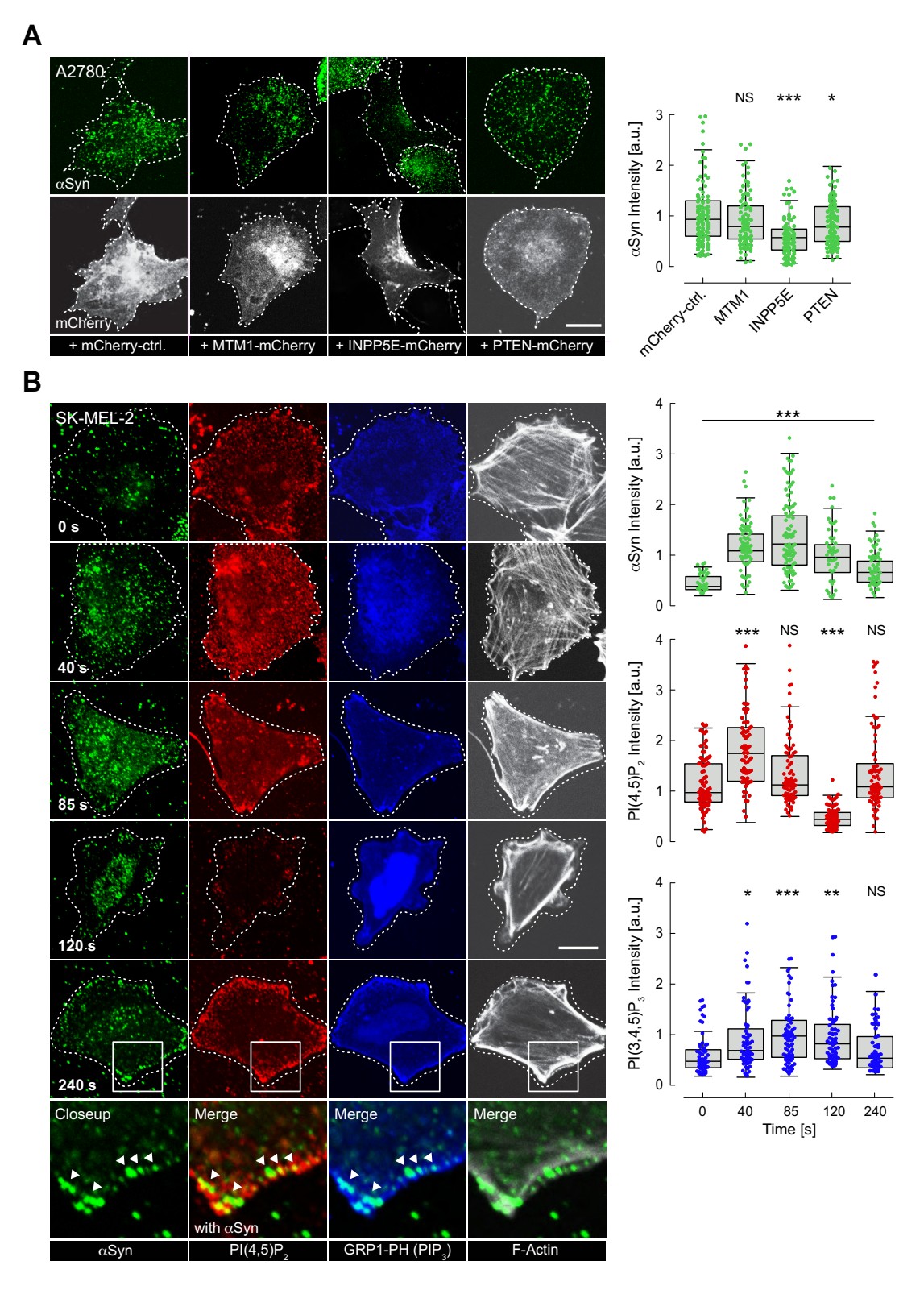

**Figure 3.** Reversible α-synuclein (αSyn)-plasma membrane (PM) localization. (**A**) Representative immunofluorescence localization of αSyn at basal A2780 PM planes by confocal microscopy. Cells transiently expressing PM-targeted, mCherry-tagged phosphatidylinositol phosphate (PIP) phosphatases, with mCherry fluorescence indicating successful transfection and phosphatase expression. Mutant, phosphatase-inactive INPP4A-mCherry-CAAX serves as the negative control (mCherry-ctrl, first panel). Scale bar is 10 μm. Box plots of αSyn immunofluorescence quantifications are shown on the

*Figure 3 continued on next page*

*Figure 3 continued*

right. Approximately 120 data points were collected per cell in four independent replicate experiments. Box dimensions represent the 25th and 75th percentiles, whiskers extend to the 5th and 95th percentiles. Data points beyond these values were considered outliers. Significance based on analysis of variance (ANOVA) tests with Bonferroni's post-tests as NS >0.05; *p<0.05; ***p<0.001. See also *Figure 3—source data 1*. (B) Time-course experiments following histamine stimulation of SK-MEL-2 cells transiently expressing histamine 1 receptor (H1R) and GRP1-PH. Immunofluorescence detection of endogenous phosphatidylinositol 4,5-bisphosphate (PIP$_2$) and αSyn by confocal microscopy of basal PM regions. GRP1-PH GFP-signals report on the presence of PIP$_3$. Phalloidin staining of F-actin marks cell boundaries. Scale bar is 10 μm. Box plots represent data points collected per cell (n ~ 80) from a single experiment, but representative of three independent experiments with similar results. Significance based on ANOVA tests with Bonferroni's post-tests as NS >0.05; *p<0.05; **p<0.01; ***p<0.001. See also *Figure 3—source data 1*.

The online version of this article includes the following source data and figure supplement(s) for figure 3:

**Source data 1.** Quantification of αSyn PM localization following PIP phosphatase expression.

**Source data 2.** Quantification of αSyn PM localization after histamine stimulation.

**Figure supplement 1.** Dynamic αSyn plasma-membrane (PM) localization upon histamine and insulin stimulation.

**Figure supplement 1—source data 1.** Quantification of αSyn PM localisation after histamine stimulation.

**Figure supplement 1—source data 2.** Uncropped western blot of IGF-1 rb expression levels in SK-MEL-2 and HEK293 cells.

**Figure supplement 1—source data 3.** Quantification of αSyn PM localization after insulin stimulation.

accommodate αSyn's helical conformation(s) (*Pranke et al., 2011*; *Pinot et al., 2014*). Thus, from a biophysical point of view, phosphatidylinositol polyphosphates satisfy many of the known requirements for efficient αSyn membrane binding. From a biological point of view, PIPs are ubiquitously expressed and stringently required for exocytosis and endocytosis, especially in neurons, where highly abundant PIP$_2$ and PIP$_3$pools (up to ~6 mol%) mark SV uptake and release sites (*James et al., 2008*). Multiple PIP-binding proteins mediate key steps in SV transmission and recycling (*Di Paolo et al., 2004*; *Milosevic et al., 2005*) and, although, αSyn has been implicated in synaptic exocytosis and endocytosis, its role(s) in these processes is ill defined (*Huang et al., 2019*).

A2780, HeLa, SH-SY5Y and SK-MEL-2 cells are poor surrogates for primary neurons and discussing our results in relation to possible scenarios at the synapse is futile. Endogenous levels of αSyn in the tested cell lines are low, particularly in comparison to presynaptic boutons, where αSyn concentrations reach up to 50 μM (*Wilhelm et al., 2014*). Similarly, the abundance of PIP$_2$ and PIP$_3$ is much smaller than at presynaptic terminals (*James et al., 2008*). Hence, αSyn-PIP scenarios in the tested cell lines and in synaptic boutons are at opposite ends of protein and lipid concentration scales. Nonetheless, we believe that key conclusions of our study are generally valid, especially because correlated localizations are equally prominent in non-neuronal cells. The affinity of αSyn to PIP$_2$ vesicles has been reported to be in the low μM range (*Narayanan and Scarlata, 2001*), similar to most other reconstituted membrane systems containing negatively charged phospholipids (*Middleton and Rhoades, 2010*; *Narayanan et al., 2005*; *Bodner et al., 2009*; *Dikiy and Eliezer, 2014*; *Fusco et al., 2014*). In comparison, average dissociation constants for canonical PIP-binding scaffolds such as PH, C2, FYVE, and ENTH domains vary between μM and mM (*Balla, 2005*; *McLaughlin et al., 2002*). By contrast, disordered polybasic PIP-binding motifs target negatively charged membranes with much weaker affinities and in a non-discriminatory fashion based on complementary electrostatic interactions (*McLaughlin and Murray, 2005*). αSyn-PIP binding may define a third class of interactions that is comparable in strength to folded protein domains, but driven, to large parts, by electrostatic contacts similar to those of polybasic motifs (*Galvagnion, 2017*). Based on these affinity considerations, we speculate that αSyn may successfully compete for cellular PIP$_2$-PIP$_3$ binding sites with other proteins, particularly when their abundance is in a comparable range. For binding scenarios at presynaptic terminals, this is likely the case.

Our findings are additionally supported by recent data showing that intracellular αSyn concentrations directly influenced cellular PIP$_2$ levels and that protein reduction diminished PIP$_2$ abundance, whereas αSyn overexpression increased PIP$_2$ synthesis and produced significantly elongated axons in primary cortical neurons (*Schechter et al., 2020a*). Conspicuously, these effects depended on αSyn's ability to interact with membranes and were absent in a membrane-binding-deficient mutant (i.e., K10E-K12E) (*Schechter et al., 2020a*). Because PM expansions require dedicated cycles of

endocytosis and exocytosis (*Pfenninger, 2009*), αSyn-PIP interactions may contribute to both types of processes, as has been suggested earlier (*Lautenschläger et al., 2017*). PM-specific αSyn-lipid interactions were additionally confirmed by 'unroofing' experiments in related SK-MEL-28 cells (*Kaur and Lee, 2020*), where endogenous protein pools colocalized with members of the exocytosis machinery including the known αSyn binding partners Rab3A (*Chen et al., 2013*) and synaptobrevin-2/VAMP2 (*Burré et al., 2010*). Two other studies implicated αSyn and αSyn-PIP$_2$ interactions in clathrin assembly and clathrin-mediated endocytosis, respectively (*Vargas et al., 2020*; *Schechter et al., 2020b*), which further strengthens the notion that phosphatidylinositol polyphosphates contribute to αSyn functions at the PM.

# Materials and methods

## Key resources table

| Reagent type (species) or resource | Designation | Source or reference | Identifiers | Additional information |
|---|---|---|---|---|
| Cell line (*Homo sapiens*) | A2780 | Sigma-Aldrich | Cat# 93112519 RRID:CVCL_0134 | |
| Cell line (*Homo sapiens*) | HeLa | Sigma-Aldrich | Cat# 93021013 RRID:CVCL_0030 | |
| Cell line (*Homo sapiens*) | SH-SY5Y | Sigma-Aldrich | Cat# 94030304 RRID:CVCL_0019 | |
| Cell line (*Homo sapiens*) | SK-MEL-2 | Dr. Ronit Sharon (Hebrew University, Israel) *Schechter et al., 2020a* | | |
| Strain, strain background (*Escherichia coli*) | BL21 (DE3) Star | Thermo Fisher Scientific | Cat# C601003 | Chemically Competent Cells |
| Antibody | Anti-αSyn (mouse monoclonal) | Santa Cruz | Cat# sc69977 RRID:AB_1118910 | IF (1:200) WB (1:100) |
| Antibody | Anti-αSyn MJFR1 (rabbit monoclonal) | Abcam | Cat# ab138501 RRID:AB_2537217 | WB (1:10,000) |
| Antibody | Anti-PI(4,5)P$_2$ (mouse monoclonal) | Echelon Biosciences | Cat# Z-P045 RRID:AB_427225 | IF (1:100) |
| Antibody | Anti-VAMP2 (rabbit monoclonal) | Cell Signalling | Cat# 13508 RRID:AB_2798240 | IF (1:200) |
| Antibody | Anti-beta actin (mouse monoclonal) | Abcam | Cat# ab6276 RRID:AB_2223210 | WB (1:5000) |
| Antibody | Anti-IGF-I Receptor ß (D23H3) (rabbit monoclonal) | Cell Signalling | Cat # 9750 RRID:AB_10950969 | WB (1:1000) |
| Antibody | Anti-mouse IgG Alexa 647 conjugated (goat polyclonal) | Abcam | Cat# ab150119 RRID:AB_2811129 | IF (1:1000) |
| Antibody | Anti-rabbit IgG Alexa 555 conjugated (donkey polyclonal) | Invitrogen | Cat# A-31572 RRID:AB_162543 | IF (1:1000) |
| Antibody | Anti-mouse IgG HRP-conjugated (goat polyclonal) | Sigma-Aldrich | Cat# A9917 RRID:AB_258476 | WB (1:10,000) |
| Antibody | Anti-rabbit IgG HRP-conjugated (goat polyclonal) | Jackson Immuno Research Laboratories | Cat# 111-035-003 RRID:AB_2313567 | WB (1:5000) |
| Recombinant DNA reagent | EGFP-PLCδ$_1$-PH | Dr. Volker Haucke (Leibniz Institute of Molecular Pharmacology, FMP-Berlin, Germany) *Várnai and Balla, 1998* | | PH domain, binds PI(4,5)P$_2$ at PM |

*Continued on next page*

*Continued*

| Reagent type (species) or resource | Designation | Source or reference | Identifiers | Additional information |
|---|---|---|---|---|
| Recombinant DNA reagent | EGFP-tagged phosphatidylinositol 4-phosphate 5-kinase type Iγ (PIPKIγ) | Dr. Volker Haucke (Leibniz Institute of Molecular Pharmacology, FMP-Berlin, Germany) *Krauss et al., 2006* | | PIP kinase, creates PI(4,5)$P_2$ at PM |
| Recombinant DNA reagent | EGFP-tagged PIPKIγ D316A (mutated) | This paper *Krauss et al., 2006* | | PIP kinase, inactive |
| Recombinant DNA reagent | EGFP-tagged PIPKIγ K188A (mutated) | This paper *Krauss et al., 2006* | | PIP kinase, inactive |
| Recombinant DNA reagent | MTM1-mCherry-CAAX | Dr. Volker Haucke (Leibniz Institute of Molecular Pharmacology, FMP-Berlin, Germany) *Posor et al., 2013* | | PIP phosphatase, acts on PI(3)P Targeted to PM |
| Recombinant DNA reagent | INPP5E-mCherry-CAAX | Dr. Volker Haucke (Leibniz Institute of Molecular Pharmacology, FMP-Berlin, Germany) *Posor et al., 2013* | | PIP phosphatase, acts on PI(4,5)$P_2$ Targeted to PM |
| Recombinant DNA reagent | PTEN-mCherry-CAAX | Dr. Volker Haucke (Leibniz Institute of Molecular Pharmacology, FMP-Berlin, Germany) *Posor et al., 2013* | | PIP phosphatase, acts on PI(3,4,5)$P_3$ Targeted to PM |
| Recombinant DNA reagent | INPP4A-mCherry-CAAX (mutated) | Dr. Volker Haucke (Leibniz Institute of Molecular Pharmacology, FMP-Berlin, Germany) *Posor et al., 2013* | | PIP phosphatase inactive Targeted to PM |
| Recombinant DNA reagent | Human histamine 1 receptor (H1R) | Dr. Ronit Sharon (Hebrew University, Israel) *Kumar et al., 2017* | | Human histamine 1 receptor |
| Recombinant DNA reagent | GRP1-PH pEGFP-C1 | Addgene *Kavran et al., 1998* | Plasmid# 71378 RRID:Addgene_71378 | PH domain binds PI(3,4,5)$P_3$ |
| Sequence-based reagent | PIPKIγ D316A_Fw | This paper | PCR primer (forward) | GTTTCAAGATCAT GGCCTACAGCCTGCTGC |
| Sequence-based reagent | PIPKIγ D316A_Rv | This paper | PCR primer (reverse) | GCAGCAGGCTGTAGG CCATGATCTTGAAAC |
| Sequence-based reagent | PIPKIγ K188A_Fw | This paper | PCR primer (forward) | GTTCATCATCGCCACC GTCATGCACAAGGAGG |
| Sequence-based reagent | PIPKIγ K188A_Rv | This paper | PCR primer (reverse) | TCGTCGTCGCTGGTGACG |
| Peptide, recombinant protein | N-terminally acetylated αSyn | This paper *Theillet et al., 2016* | | Purified from *E. coli* BL21 (DE3) Star |
| Peptide, recombinant protein | N-terminally truncated (ΔN) αSyn | This paper *Theillet et al., 2016* | | Purified from *E. coli* BL21 (DE3) Star |
| Peptide, recombinant protein | N-terminally acetylated αSyn (F4A-Y39A) mutated | This paper *Theillet et al., 2016* | | Purified from *E. coli* BL21 (DE3) Star |
| Commercial assay or kit | Q5 Site-Directed Mutagenesis Kit | New England BioLabs | Cat# E0554S | |

*Continued on next page*

*Continued*

| Reagent type (species) or resource | Designation | Source or reference | Identifiers | Additional information |
|---|---|---|---|---|
| Commercial assay or kit | BCA protein quantification kit | Thermo Fisher | Cat# 23227 | |
| Commercial assay or kit | SuperSignal West Pico PLUS Chemiluminescent Substrate | Thermo Fisher | Cat# 34579 | |
| Chemical compound, drug | All-trans retinoic acid | Sigma-Aldrich | Cat# R2625 | |
| Chemical compound, drug | Recombinant human/murine/rat BDNF | Peprotech | Cat# 450-02 | |
| Software, algorithm | Image Analysis FIJI | imagej.net/Fiji *Schindelin et al., 2012* | RRID:SCR_002285 | |
| Software, algorithm | Multi-dimensional NMR data processing PROSA | Dr. Peter Güntert Goethe-University Frankfurt am Main, Germany *Güntert et al., 1992* | | |
| Software, algorithm | Computer-aided NMR resonance assignment CARA | cara.nmr.ch | | PhD thesis Rochus Keller ETH Nr. 15947 |
| Others | Lipofectamine 3000 | Thermo Fisher Scientific | Cat# L3000015 | |
| Others | TransIT-X2 | Mirus Bio | Cat# MIR 6000 | |
| Others | DOPC | Avanti Polar Lipids | Cat# 850375 | |
| Others | Brain PI(4,5)P$_2$ | Avanti Polar Lipids | Cat# 840046 | |
| Others | IP$_6$ | Dr. Dorothea Fiedler (Leibniz Institute of Molecular Pharmacology, FMP-Berlin, Germany) | | In-house synthesis |
| Others | PLC from *Clostridium perfringens* (*Clostridium welchii*) | Sigma-Aldrich | Cat# 9001-86-9 | |

## Mammalian cell lines and growth media

All cells lines used in this study are described in the Key resources table. Cells were grown in humidified 5% (v/v) $CO_2$ incubators at 37 °C in the following media supplemented with 10% (v/v) fetal bovine serum: RPMI 1640 (A2780), low glucose Dulbecco's modified eagle medium (DMEM)(HeLa), DMEM-Ham's F-12 (SH-SY5Y), and minimum essential medium (MEM) with 1% non-essential amino acids and 2 mM glutamine (SK-MEL-2). Cells were split at 70–80% confluence with a passage number below 20 for all experiments. All cell lines were routinely tested for being mycoplasma free.

## Transient cell transfections

A2780 cells were seeded on fibronectin (Sigma-Aldrich, USA) coated 18 mm cover slips in 12-well plates at a density of $3 \times 10^5$ cells. Cells were transfected using Lipofectamine 3000 (Thermo Fisher, USA) according to the manufacturer's instructions. Undifferentiated SH-SY5Y and SK-MEL-2 cells were seeded on 18 mm coverslips at a density of $2 \times 10^5$ cells and transfected using TransIT-X2 (Mirus Bio, USA) according to the manufacturer's instructions. Details of transfection plasmids and mutagenesis primers are provided in the Key Resources Table. Kinase-inactive PIPKIγ mutants were generated with the Q5 site-directed mutagenesis kit (New England BioLabs, USA). Mutant PIPKIγ was confirmed by DNA sequencing. 1 μg of plasmids was used in all cases. Following transfection, cells were grown for 24 hr before analysis.

## siRNA knockdown experiments

Commercial siRNA mixtures against human αSyn (Dharmacon, USA, ON-TARGET plus human SNCA, cat.# L-002000-00-0005) and a non-targeted control (cat.# D-001810-10-05) were used. A2780 cells were seeded at a density of $6 \times 10^5$ cells and transfected with 1.7 µg of the respective siRNA mixtures using Lipofectamine 3000 according to the manufacturer's instructions. After transfection, cells were grown for 48 hr before analysis.

## SH-SY5Y differentiation

Stringent SH-SY5Y differentiation was performed according to *Encinas et al., 2000*. In short, cells were seeded at a density of $2 \times 10^5$ in collagen-coated six-well plates. Twenty-four hours after seeding, cells were pre-differentiated with 10 µM of all trans RA in growth medium for 5 days. Subsequently, cells were cultured in serum-free medium supplemented with 50 ng/mL BDNF (Peprotech, Israel) for 7 days to obtain terminal neuronal differentiation. During the entire process, growth media were exchanged every 2–3 days.

## Immunofluorescence

For IF imaging of endogenous αSyn and expressed PIP-kinase/phosphatases, cells were washed 3 × 5 min with PBS and fixed in 4% (w/v) paraformaldehyde (PFA) for 15 min at room temperature (RT). For PM staining with 5 µg/mL Alexa Fluor 350/tetramethylrhodamine conjugated to wheat germ agglutinin (WGA) (Invitrogen, USA), cells were fixed and washed with PBS before application for 10 min at RT. Excess dye was washed off with PBS. For antibody staining, cells were permeabilized with 0.5% saponin in PBS for 10 min and blocked with 5% (w/v) bovine serum albumin (Sigma-Aldrich, USA) in PBS for 30 min. After blocking, cells were incubated with anti-αSyn antibody for 90 min at RT. After washing 3 × 5 min with PBS, cover slips were incubated with Alexa Fluor-tagged secondary antibody for 45 min at RT. Before confocal microscopy, cover slips were mounted with Immu-Mount (Thermo Fisher, USA), after 3 × 5 min PBS washes. IF detection of PI(4,5)P$_2$ at the PM was performed according to *Hammond et al., 2009*) with slight modifications. A2780 and SK-MEL-2 cells were cultured on fibronectin-coated coverslips and pre-extracted in PHEM buffer (60 mM PIPES (piperazine-N,N′-bis(2-ethanesulfonic acid), 25 mM HEPES (4-(2-hydroxyethyl)-1-piperazineethanesulfonic acid), 5 mM EGTA (ethylene glycol-bis(β-aminoethyl ether)-N,N,N′,N′-tetraacetic acid), 1 mM MgCl$_2$) to remove the majority of soluble cytoplasmic proteins. Cells were fixed with 4% PFA and 0.2% glutaraldehyde in PHEM buffer for 15 min at RT. All post-fixation steps until mounting were carried out at 4 ˚C. Washes were performed with ice-cold PIPES buffer (20 mM PIPES, pH 6.8, 137 mM NaCl, 2.7 mM KCl) to minimize damage to endogenous PIP moieties. Following fixation, cells were washed thrice in PIPES buffer containing 50 mM NH$_4$Cl and subsequently blocked and permeabilized in PIPES buffer supplemented with 5% 'normal goat serum' and 0.5% saponin for 30 min. Post blocking, cells were incubated with anti-PI(4,5)P$_2$ and anti-αSyn antibodies for 60 min, washed thrice, and incubated with Alexa Fluor 647 secondary antibody for 45 min. Before confocal microscopy, cover slips were mounted with Immu-Mount (Thermo Fisher, USA) after 3 × 5 min PIPES buffer washes. All primary and secondary antibody details are provided in the Key Resources Table.

## Confocal microscopy

Confocal microscopy imaging was performed on a Nikon spinning disk confocal microscope with an oil ×60 objective and additional ×1.5 magnification. Four channels in five optical sections from the basal PM plane were acquired with excitation wavelengths of 405 (blue, 50% laser power, for WGA), 488 (green, 20% for GFP), 568 (red, 20% for mCherry), and 647 (far-red, 20% for goat anti-mouse) with 200 ms exposure times. At least 25 images per biological replicate were collected and 3–4 replicates per experiment were analyzed.

## TIRF microscopy

For TIRF localization of endogenous αSyn at the PM, A2780, HeLa, SH-SY5Y, and SK-MEL-2 cells were cultured on 18 mm fibronectin-coated coverslips at a density of $2 \times 10^5$ cells for 24 hr and fixed with 4% PFA. After fixation, antibody detection was performed as described in the previous section. Coverslips for TIRF imaging were mounted in PBS after immunostaining and imaged on an Andor Dragonfly spinning disk microscope with a TIRF 100×/NA 1.45 oil objective. For TIRF

detection of PM-proximal fluorescence signals, evanescent fields were kept at 50 nm in all experiments. Four lasers operating at 405 nm (15% laser power), 488 nm (20% laser power), 561 nm (20% laser power), and 647 nm (20% laser power) were used for fluorophore excitation along with 200 ms exposure time for image acquisition. At least 20 images per biological replicate were collected and three replicates per experiment were analyzed.

## Histamine and insulin stimulation

PI-3 kinase activity was stimulated by either insulin or histamine addition. For insulin stimulation via the endogenously expressed insulin-like growth factor-1 receptor β (IGF-1 Rβ) (*Dricu et al., 1999*), SK-MEL-2 cells were seeded on coverslips and starved in Hank's balanced salt solution (HBSS) for 18 hr, as described (*Gray et al., 1999*). 100 nM of insulin was added for 10 min and cells were fixed immediately afterwards. For histamine stimulation, SK-MEL-2 cells were seeded on 18 mm coverslips at a density of $2 \times 10^5$, transiently transfected with histamine 1 receptor (H1R) and serum-starved for 3 hr, as described in *Mizuguchi et al., 2011*. 500 µM of histamine was added and cells were fixed at indicated time points. All cell samples were further processed as previously outlined for TIRF procedures. F-Actin was detected by Phalloidin-Alexa Fluor 405 staining (1:400, Invitrogen, USA) during secondary antibody incubation.

## Image analysis and quantification

Image analysis and quantification were performed in Fiji (*Schindelin et al., 2012*). For confocal image quantification, focal planes of apical and basal PMs were selected manually. Images were segmented based on GFP signals by automatic thresholding according to *Huang and Wang, 1995*. Threshold regions were marked as regions of interest (ROIs), copied to the far-red channel (αSyn IF), and fluorescence intensities were determined. In the box plots of *Figure 1E*, *Figure 1—figure supplement 2A* and *Figure 3A, B*, each ROI corresponds to a single cell and is represented as a data point. For TIRF data in *Figure 3—figure supplement 1B, D*, images were segmented based on Phalloidin signals via automated thresholding using the default algorithm in Fiji (*Zonderland et al., 2019*). Different than for confocal images in *Figures 1* and *3*, TIRF ROIs consist of multiple adjacent cells in a single frame that were copied to the far-red channel (αSyn IF). ROIs of less than 2 µm² in size were excluded. The Fiji particle counting routine was used to determine the number of αSyn puncta in each ROI. The number of cells in each image was determined manually based on cell outlines marked by Phalloidin. In *Figure 3—figure supplement 1B, D*, data points in box plots were calculated by dividing the number of αSyn puncta per image by the cell count. All box plots depict median values (center lines) with box dimensions representing the 25th and 75th percentiles. Whiskers extend to 1.5 times the interquartile range and depict the 5th and 95th percentiles. Each box plot in *Figure 1E*, *Figure 1—figure supplement 2A* and *Figure 3A* corresponds to 110–120 data points combined from four independent biological replicates. Box plots in *Figure 3B* contain data points collected per cell (n ~ 80) from a single experiment but representative of three independent experiments with similar results. Box plots in *Figure 3—figure supplement 1B, D* contain data points from ~120 cells, combined from three independent biological replicates.

## Statistical analysis

For box plots, data points considered 'outliers' were determined based on the criteria defined in the *Grubbs, 1969* outlier test and omitted. Analysis of variance tests with Bonferroni's post-tests (*Armstrong, 2014*; *Dunn, 1961*) were used to determine the statistical significance of experiments with more than two samples, whereas Student's *t* tests were performed to assess statistical differences between samples (*Kalpić et al., 2011*). Significance is given as NS >0.05; *p<0.05; **p<0.01; ***p<0.001. Absolute p values are given in the respective Source data files.

## Cell lysate preparation

Lysates of A2780, HeLa, SH-SY5Y, and SK-MEL-2 cell lines were prepared by detaching ~5–10 million cells with trypsin/EDTA (0.05%/0.02%) and harvested by centrifugation at 130 × *g* for 5 min at 25 ℃. Sedimented cells were washed once with PBS, counted on a haemo-cytometer, and pelleted again by centrifugation. After resuspending cells in PBS with proteinase inhibitor cocktail (Roche, Switzerland), yielding a cell count of $2 \times 10^7$ cells/mL, they were lysed by repeated freeze–

thaw cycles. Lysates were cleared by centrifugation at 16,000 × $g$ for 30 min. Supernatants were removed, and total protein concentration was measured with a bicinchoninic acid (BCA) assay kit (Thermo Fisher, USA). For western blotting, 25 µg of protein (per lane, *Figure 1—figure supplement 1C*) or 50 µg of protein (per lane, all other figures) were applied onto SDS-PAGE.

## Western blotting

Cell lysates and recombinant protein samples were boiled in Laemmli buffer for 10 min before SDS-PAGE separation on commercial, precast 4–18% gradient gels (BioRad, USA). Recombinant N-terminally acetylated α-, β-, and γ-Syn were loaded as reference inputs at specified concentrations (see 'Recombinant protein expression and purification'). Proteins were transferred onto polyvinylidene fluoride (PVDF) membranes and fixed with 4% (w/v) PFA in phosphate-buffered saline (PBS) for 1 hr (*Lee and Kamitani, 2011*). Membranes were washed 2× with PBS, 2× with tris-buffered saline with 0.1% tween 20 (TBST), and blocked in 5% milk-TBST for 1 hr. After blocking, blots were incubated with primary antibodies overnight at 4 °C. Membranes were washed and probed with horseradish peroxidase (HRP)-conjugated secondary antibodies for 1 hr. All antibody details including respective dilutions are provided in the Key Resource Table. Membranes were developed using the SuperSignal West Pico Plus reagent (Thermo Fisher, USA), and luminescence signals were detected on a BioRad Molecular Imager.

## Western blot quantification

Intensities of αSyn and β-actin bands were quantified using the ImageLab software (BioRad, USA). αSyn reference input was used to generate a standard curve. For cell lysate samples, αSyn intensity was normalized according to the β-actin signal and cell lysate concentrations were calculated with respect to αSyn standards. Error bars denote background (noise).

## Recombinant protein expression and purification

$^{15}$N isotope-labeled, N-terminally acetylated, human αSyn was produced by co-expressing PT7-7 plasmids with yeast N-acetyltransferase complex B (NatB) (*Johnson et al., 2010*) in *Escherichia coli* BL21 Star (DE3) cells in M9 minimal medium supplemented with 0.5 g/L of $^{15}$NH$_4$Cl (Sigma-Aldrich, USA). Unlabeled α-, β- and γ-Syn were produced in Luria-Bertani (LB) medium. Generation of αSyn mutants ΔN and F4A-Y39A was described previously (*Theillet et al., 2016*). For recombinant protein purification under non-denaturing conditions, we followed the protocol by *Theillet et al., 2016*. Purification of αSyn F4A-Y39A was identical to wild-type αSyn. Lacking the N-terminal substrate specificity for NatB, αSyn ΔN was produced in its non-acetylated form and purified as the wild-type protein. Methionine-oxidized $^{15}$N isotope-labeled wild-type αSyn was expressed and purified as described (*Binolfi et al., 2016*). Protein samples were concentrated to 1–1.2 mM in NMR buffer (25 mM sodium phosphate, 150 mM NaCl) at pH 7.0. Protein concentrations were determined spectrophotometrically by UV absorbance measurements at 280 nm with ε = 5960 M$^{-1}$ cm$^{-1}$ for α-, β-Syn ΔN, and methionine-oxidized αSyn. For αSyn F4A-Y39A and γ-Syn, ε = 4470 and 1490 M$^{-1}$ cm$^{-1}$ were used. Final aliquots of protein stock solutions were snap frozen in liquid nitrogen and stored at −80 °C until use.

## Reconstituted PI(4,5)P$_2$ vesicles

Phospholipids were purchased from Avanti Polar Lipids (USA). Large unilamellar vesicles (LUVs, 100 nm) were prepared from 100% brain (porcine) PIP$_2$. A thin lipid film was formed in a glass vial by gently drying 1 mg of PIP$_2$ in chloroform-methanol under a stream of nitrogen. To remove residual traces of organic solvent, the lipid film was placed under vacuum overnight. 0.5 mL NMR buffer was then added to hydrate the lipid film for 1 hr at RT while agitating. After five freeze–thaw cycles on dry ice and incubation in a water bath at RT, the lipid suspension was sonicated at 4 °C for 20 min at 30% power settings (Bandelin, Germany). Resulting PIP$_2$ LUVs (2 mg/mL) were used immediately. αSyn:PIP$_2$ molar ratios for sample preparations were calculated using a PIP$_2$ lipid mass of 1096 Da. For PIP$_2$ titration experiments, 1-, 5-, 10-, 15-, 20-, and 30-fold molar excess of lipids was added to 60 µM of $^{15}$N isotope-labeled, N-terminally acetylated αSyn (total volume 120 µL) and αSyn PIP$_2$ samples were incubated for 45 min at RT before NMR and CD measurements. Following the same procedure, mixed PC-PIP (PC:PIP$_2$ mass ratio 9:1) suspensions were prepared using 9 mg of 1,2-

dioleoyl-sn-glycero-3-phosphocholine (DOPC, 786 Da) and 1 mg $PIP_2$. The dried lipid film was hydrated with 0.25 mL NMR buffer. The $PC-PIP_2$ suspension was then extruded through polycarbonate membranes with a pore size of 100 nm according to the manufacturer's instructions (mini-extruder, Avanti Polar Lipids) and resulting $PC-PIP_2$ LUVs (40 mg/mL, $PC:PIP_2$ molar ratio 13:1) were used immediately. For sample preparations, an approximate $PC-PIP_2$ lipid mass of ~810 Da ((13/14) × 786 Da + (1/14) × 1096 Da) was used to calculate the $\alpha Syn:PC-PIP_2$ molar ratios. $^{15}N$ isotope-labeled, N-terminally acetylated $\alpha Syn$ (60 µM) was incubated with ~80-, ~170-, ~340-, and ~680-fold molar excess of total $PC-PIP_2$ lipids. $\alpha Syn$ $PC-PIP_2$ samples (total volume 120 µL) were incubated for 45 min at RT before CD, NMR, and DLS experiments. Synthetic inositol hexaphosphate ($IP_6$) was kindly provided by Dr. Dorothea Fiedler, Department of Chemical Biology, Leibniz Institute of Molecular Pharmacology (FMP-Berlin). Before NMR measurements, 50 µM $\alpha Syn$ was incubated with 200 µM $IP_6$ in NMR buffer (total volume 120 µL) for 45 min at RT.

## Phospholipase C reaction

PLC was purchased from Sigma-Aldrich (USA) and the lyophilized powder was dissolved in NMR buffer at 1000 units (U)/mL. $\alpha Syn$ $PC-PIP_2$ samples at ~680-fold molar excess of $PC-PIP_2$ lipids (60 µM $\alpha Syn$, ~40 mM $PC-PIP_2$) were incubated while agitating at 37 °C for 45 min with 10 U of PLC and 1 mM phenylmethylsulfonyl fluoride (PMSF) in a total volume of 120 µL.

## NMR spectroscopy

For best comparison of protein reference and $\alpha Syn$-lipid NMR data, final concentrations of $^{15}N$ isotope-labeled, N-terminally acetylated $\alpha Syn$ samples were adjusted to 60 µM, supplemented with 5% $D_2O$, and measured in 3 mm (diameter) Shigemi tubes in all cases. NMR experiments were acquired on a Bruker 600 MHz Avance spectrometer equipped with a cryogenically cooled proton-optimized $^1H\{^{13}C/^{15}N\}$ TCI probe. Reference and $\alpha Syn$-lipid NMR spectra were acquired with identical spectrometer settings and general acquisition parameters. Specifically, we employed 2D $^1H$-$^{15}N$ SOFAST HMQC NMR pulse sequences (*Schanda et al., 2005*) with 512 x 128 complex points for a sweep width of 28.0 ppm ($^{15}N$) and 16.7 ppm ($^1H$), 128 scans, 60 ms recycling delay at 283 K. Inspection of the highly pH-sensitive His50 (H50) $^1H$-$^{15}N$ chemical shift indicated that the sample pH changed from 7 to 6.5 during the PLC reaction (*Figure 2E*). To accurately delineate $I/I_0$ values, we recorded reference NMR spectra at pH 6.5. All NMR spectra were processed with PROSA, zero-filled to four times the number of real points and processed without window function. Visualization and data analysis were carried out in CARA. NMR signal intensity ratios ($I/I_0$) of isolated $\alpha Syn$ ($I_0$) and in the presence of lipids ($I$) were determined for each residue by extracting maximal signal peak heights in the respective 2D $^1H$-$^{15}N$ NMR spectra.

## CD spectroscopy

NMR samples of isolated $\alpha Syn$ and $\alpha Syn$ in the presence of lipid vesicles were diluted with NMR buffer to a final protein concentration of 10 µM for CD measurements. CD spectra (200–250 nm) were collected on a Jasco J-720 CD spectropolarimeter in a 1 mm quartz cell at 25 °C. One replicate per sample was recorded. Six scans were averaged and blank samples (without $\alpha Syn$) were subtracted from protein spectra to calculate the mean residue weight ellipticity ($\theta_{MRW}$).

## Dynamic light scattering

DLS measurements were acquired on a Zetasizer Nano ZS (Malvern Instruments, UK) operating at a laser wavelength of 633 nm equipped with a Peltier temperature controller set to 25 °C. Data were collected on all NMR samples containing $\alpha Syn$, isolated $PC-PIP_2$ vesicles and $PC-PIP_2$ vesicles in the presence of $Ca^{2+}$ and PLC. Using the Malvern DTS software, mean hydrodynamic diameters were calculated from three replicates of the same sample in the intensity-weighted mode.

## Negative-stain electron microscopy

NMR samples of $\alpha Syn$ at ~30- and ~680-fold molar excess of $PIP_2$ and $PC-PIP_2$ lipids were diluted to a protein concentration of ~10 µM in NMR buffer. 5 µL aliquots were added to glow-discharged carbon-coated copper grids for 1 min. Excess liquid was removed with filter paper and grids were

washed twice with $H_2O$ before staining with 2% (w/v) uranyl acetate for 15 s. Negative-stain transmission EM images were acquired on a Technai G2 TEM.

## Acknowledgements

We are grateful to Dr. Peter Schmieder and Monika Beerbaum for excellent maintenance of the NMR infrastructure at the Leibniz Institute of Molecular Pharmacology (FMP Berlin) and Dr. Tali Scherf for NMR infrastructure maintenance at the Weizmann Institute of Science. We thank Dr. Dmytro Puchkov (FMP-Berlin) for assistance with negative-stain electron microscopy and Drs. Michael Krauss and Volker Haucke (FMP-Berlin) for tools and reagents, helpful discussions throughout the project and useful feedback on the manuscript. Dr. Dorothea Fiedler (FMP-Berlin) for sharing aliquots of $IP_6$. We also thank Drs. Martin Lehmann (Cellular imaging, FMP-Berlin) and Yoseph Addadi (Life Sciences Core Facilities, Weizmann Institute of Science) for excellent maintenance of imaging facilities and their support at the respective institutes. We acknowledge the highly valuable input by Drs. Meir Schechter and Ronit Sharon, Hebrew University Jerusalem, especially with regard to time-resolved histamine experiments. We further thank them for kindly providing aliquots of the SK-MEL-2 cell line. We are grateful to Drs. Ori Avinoam, Hagen Hofmann (Weizmann), and Andres Binolfi (CONICET) for carefully reading the manuscript. CE was supported by a Swiss National Science Foundation (SNSF) Advanced -Postdoc.Mobility fellowship P300PA_160979. PS acknowledges funding by the European Research Council (ERC) Consolidator Grant NeuroInCellNMR (647474). TIRF imaging was made possible with the help and support of the de Picciotto Cancer Cell Observatory in memory of Wolfgang and Ruth Lesser. Work in the Selenko laboratory is supported by the Willner Family Foundation.

## Additional information

### Funding

| Funder | Grant reference number | Author |
| --- | --- | --- |
| European Research Council | 647474 | Philipp Selenko |
| Swiss National Science Foundation | P300PA_160979 | Cedric Eichmann |

The funders had no role in study design, data collection and interpretation, or the decision to submit the work for publication.

### Author contributions

Reeba Susan Jacob, Conceptualization, Resources, Formal analysis, Supervision, Funding acquisition, Validation, Investigation, Visualization, Methodology, Writing - original draft, Project administration, Writing - review and editing; Cédric Eichmann, Alessandro Dema, Conceptualization, Resources, Formal analysis, Validation, Investigation, Visualization, Methodology, Writing - original draft, Writing - review and editing; Davide Mercadante, Conceptualization, Resources, Formal analysis, Validation, Investigation, Visualization, Methodology, Writing - review and editing; Philipp Selenko, Conceptualization, Resources, Formal analysis, Supervision, Funding acquisition, Validation, Investigation, Methodology, Writing - original draft, Project administration, Writing - review and editing

### Author ORCIDs

Reeba Susan Jacob https://orcid.org/0000-0002-0676-834X
Cédric Eichmann https://orcid.org/0000-0002-8476-1936
Alessandro Dema https://orcid.org/0000-0003-0976-9396
Davide Mercadante https://orcid.org/0000-0001-6792-7706
Philipp Selenko https://orcid.org/0000-0002-2590-5899

### Decision letter and Author response

Decision letter https://doi.org/10.7554/eLife.61951.sa1
Author response https://doi.org/10.7554/eLife.61951.sa2

## Additional files

### Supplementary files

• Transparent reporting form

### Data availability

All data generated or analysed during this study are included in the manuscript and supporting files. Source data files have been provided for Figures 1,2,3 and all figure supplements.

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
