## [Decision Letter]

**Acceptance summary:**

In this short report it is shown that phosphatidylinositol 4,5-bisphosphate (PIP2) and phosphatidylinositol 3,4,5-trisphosphate (PIP3), two highly acidic components of inner plasma membrane leaflets, mediate plasma membrane localization of endogenous pools of α-synuclein in A2780, HeLa, SH-SY5Y and SK-MEL-2 cells. Moreover, it is shown that PIP2 synthesizing kinases and hydrolyzing phosphatases can redistribute α-synclein in cells. In addition, structural details of the interactions of αSyn with PIP2 containing membranes have been revealed by NMR spectroscopy.

**Decision letter after peer review:**

Thank you for submitting your article "α-Synuclein plasma membrane localization correlates with cellular phosphatidylinositol polyphosphate levels" for consideration by *eLife*. Your article has been reviewed by three peer reviewers, one of whom is a member of our Board of Reviewing Editors, and the evaluation has been overseen by Olga Boudker as the Senior Editor. The following individual involved in review of your submission has agreed to reveal their identity: David Eliezer (Reviewer #2).

The reviewers have discussed the reviews with one another and the Reviewing Editor has drafted this decision to help you prepare a revised submission.

Summary:

The normal behavior and function of α-synuclein remain poorly understood, at least in part because the majority of studies of this protein focus on its pathological self-assembly and deposition, processes that are thought to be critical for its role in neurodegeneration. Whether or not the latter is true, an improved understanding of the cell biology of α-synuclein is necessary to clarify when and how this protein deviates from its normal physiological contexts during the course of disease. To that end, understanding the cellular localization of the protein, (which has remained highly controversial) and to explore how it is regulated (which has hardly been studied) are important goals that are addressed by this study. Given the importance of PIPs in membrane trafficking pathways, and the growing consensus that α-synuclein functions in the arena of synaptic vesicle trafficking, uncovering a role for PIPs in attracting α-synuclein to membranes is a significant contribution to our understanding of this protein.

In this short report it is shown that phosphatidylinositol 4,5-bisphosphate (PIP2) and phosphatidylinositol 3,4,5-trisphosphate (PIP3), two highly acidic components of inner PM leaflets, mediate plasma membrane localization of endogenous pools of αSyn in A2780, HeLa, SH-SY5Y and SK-MEL-2 cells. Although it is already known that αSyn binds to PIP2 and other anionic lipids and that this binding induces α-helicity in αSyn this work additionally shows that PIP2 synthesizing kinases and hydrolyzing phosphatases can redistribute αSyn in cells. Moreover, structural details of the interactions of αSyn with PIP2 containing membranes have been revealed by NMR spectroscopy.

Essential revisions:

While overall, there is good evidence that αSyn targets PIP2 at the plasma membrane, the in vivo relevance remains unclear, in particular with neuronal activity-dependent roles of PIP2 in both, synaptic vesicle exo- and endocytosis, and of PIP3 in postsynaptic receptor trafficking. The reviewers are concerned about the cellular systems that were chosen to assess αSyn plasma membrane localization – mostly of non-neuronal origin – which the authors also highlight themselves. The authors have the neuronal SH-SY5Y cell line and produced little data with this line, but most of the experiments presented here were performed in A2780 (ovarian carcinoma), HeLa (cervical cancer) and SK-MEL-2 (skin metastasis) cells. If some of the experiments could be repeated in the neuroblastoma cell line, ideally differentiated into a more neuron-like shape, it would make the paper more compelling.

Figure 1E and Figure 1—figure supplement 1B: The authors describe increased amounts of αSyn at the PM by expression of PIPKIγ. Are total αSyn levels affected, and could this result in an increased pool on the PM? Please test by western blotting.

Figure 1—figure supplement 1D: An overlay or quantification of the localization of αSyn at the PM would be helpful. The overlap in A2780 cells seems to be high, but the overlap in SH-SY5Y and SK-MEL-2 cells is not particularly convincing.

Figure 2C: Please justify the use of 100 nm vesicles. If the PM should be mimicked, the vesicles should have a larger diameter.

Figure 3A: The quantification of αSyn intensity with expression of INPP5E is not reflected by the provided image where there is essentially no αSyn signal. Please provide a more representative image.

Figure 3A: The authors argue that αSyn has a specificity for PIP2 and PIP3, but why then are there no apparent changes between αSyn intensity upon expression of MTM1 and INPP5E (which result in production of π and PI4P, respectively) versus PTEN (which produces PIP2 from PIP3)? In addition, the authors describe that "only the conversion of PIP2 to PI4P by INPP5E led to a marked reduction" but their statistics show significant changes for all three phosphatases compared to mCherry only.

Figure 3: There may be a contribution of coincidental labeling. Could the authors reduce PIP2 and/or PIP3 levels as in panel 3A and repeat their analysis?

Are the observed CD spectra and NMR spectra shown in Figure 2 consistent with residues ~ 1-100 in an helical conformation and the remaining ~ 40 residues being unstructured under high PIP2 to αSyn ratios, similar to the solution NMR structure of micelle-bound αSyn (PDB ID 1XQ8) or similar to the solid state NMR studies of αSyn interactions with lipid bilayers by Fuso et al., Nature Comm, 2014? In other words, are residues 100-140 unstructured in all these studies? Please comment.

The experiments with calcium (Figure 2E) suggest that the first ten residues always interact with PC-PIP2 vesicles, although overall binding was greatly reduced. However, plasma membranes contain anionic lipids that may mitigate the effect of calcium that is observed in these experiments. Please comment.

---

## [Author Response]

Essential revisions:While overall, there is good evidence that αSyn targets PIP2 at the plasma membrane, the in vivo relevance remains unclear, in particular with neuronal activity-dependent roles of PIP2 in both, synaptic vesicle exo- and endocytosis, and of PIP3 in postsynaptic receptor trafficking. The reviewers are concerned about the cellular systems that were chosen to assess αSyn plasma membrane localization – mostly of non-neuronal origin – which the authors also highlight themselves. The authors have the neuronal SH-SY5Y cell line and produced little data with this line, but most of the experiments presented here were performed in A2780 (ovarian carcinoma), HeLa (cervical cancer) and SK-MEL-2 (skin metastasis) cells. If some of the experiments could be repeated in the neuroblastoma cell line, ideally differentiated into a more neuron-like shape, it would make the paper more compelling.

We have now included new figure panels depicting co-localization of endogenous αSyn with PIP2 in fully differentiated SH-SY5Y cells. Specifically, we followed a 14-day differentiation protocol that employs initial stimulation with retinoic acid (RA) for 5 days, followed by brain-derived neurotrophic factor (BDNF) treatment for 7 additional days. The procedure commonly accepted as the most stringent differentiation routine for SH-SY5Y cells^1^, especially in comparison to differentiation with RA only^2^. In line with expected results, we observed substantial neurite outgrowth and shrinkage of cell bodies, concomitant with increased expression of endogenous αSyn (new Figure 1—figure supplement 1C). We detected αSyn in cell bodies and neurites, although protein pools were particularly abundant along neurites where they colocalize with PIP2 in expanded structures reminiscent of synaptic boutons (new Figure 1C, D, lower panel). Indeed, co-staining for the vesicular SNARE-component synaptobrevin-2 (Syb2/VAMP2) confirmed that αSynPIP2 and αSyn-Syb2/VAMP2 localize to these structures (new Figure 1—figure supplement 1D, E). As the reviewers are obviously aware, Syb2/VAMP2 is one of the few “broadly accepted” binding partners of αSyn in presynaptic terminals^3^.

To the best of our knowledge, these are the first reported results of endogenous αSyn-PIP2 and Syb2/VAMP colocalization in fully differentiated SH-SY5Y cells.

We updated the manuscript accordingly:

Abstract: “… endogenous pools of αSyn in A2780, HeLa, SK-MEL-2 and differentiated and undifferentiated neuronal SH-SY5Y cells.”

Results: “We verified PM colocalization of αSyn with PIP_2_ in SH-SY5Y cells that we differentiated into dopaminergic-like neurons following a stringent protocol and stimulation with retinoic acid (RA) and brain-derived neurotrophic factor (BDNF)^1^ (Figure 1—figure supplement 1C). We found prominent pools of αSyn in expanded structures reminiscent of synaptic boutons along neurites, where they colocalized with PIP_2_ (Figure 1C, bottom panel, Figure 1D and Figure 1—figure supplement 1D). These structures also stained positive for the presynaptic V-SNARE component syanptobrevin2/VAMP2, a known binding partner of αSyn^3^ (Figure 1—figure supplement 1E).”

We also attempted to transfect fully differentiated SH-SY5Y cells with GFP-PIPKIγ to ask whether ectopic expression of the PIP2 kinase triggered localization of higher levels of endogenous αSyn at the PM. This is a challenging experiment in any kind of terminally differentiated cell line. We accomplished this goal after careful evaluation of various transfection agents and procedures. However, we were unable to perform consistent confocal imaging at equivalent planes for cell bodies and neurites given their highly irregular shapes and extended morphologies. Accordingly, we did not quantify αSynfluorescence on a per cell basis in *differentiated SH-SY5Y cells*, as we had done for A2780 and HeLa cells.

Instead, we extended the analysis of induced αSyn PM localization upon PIPKIγ expression in *undifferentiated SH-SY5Y cells*, which is now included in the new Figure 1E (right panel). Compared to the original submission, we increased the number of analyzed cells (n > 120) to derive quantitative information about the significance of the observed effect. We are happy to report that we obtained *p*-values < 0.001 (***) for this and all other cell lines indicating the general significance and reproducibility of the PMtargeting effect upon induced accumulation of PIP2 at the PM. We moved the analysis of HeLa cells to Figure 1—figure supplement 2A.

We updated the manuscript accordingly (see also the next question).

Results: “We obtained similar results in undifferentiated SH-SY5Y and HeLa cells (Figure 1E and Figure 1—figure supplement 2A) and confirmed that transient

PIPKIγ expression did not affect overall αSyn abundance (Figure 1—figure supplement 2B).”

Figure 1E and Figure 1—figure supplement 1B: The authors describe increased amounts of αSyn at the PM by expression of PIPKIγ. Are total αSyn levels affected, and could this result in an increased pool on the PM? Please test by western blotting.

We quantified levels of endogenous αSyn in A2780 and undifferentiated SHSY5Y cells upon expression of wild-type (WT) GFP-PIPKIγ and GFP only. In addition, we included two PIPKIγ mutants (D316A and K188A), which are catalytically inactive but targeted to the PM similar to WT PIPKIγ. As can be appreciated from the Western blots (new Figure 1 —figure supplement 2B), transient expression of any of these constructs did not alter the global levels of endogenous αSyn. We include this analysis in the revised manuscript and updated the text accordingly (see above).

Figure 1—figure supplement 1D: An overlay or quantification of the localization of αSyn at the PM would be helpful. The overlap in A2780 cells seems to be high, but the overlap in SH-SY5Y and SK-MEL-2 cells is not particularly convincing.

There appears to be a misunderstanding as to what the PM-dye in these images shows. This panel confirms that the chosen TIRF fields correctly represent the PM, which is identified by staining with tetramethylrhodamine-WGA (as specified in the figure legend). No co-localization between this dye and αSyn is expected. The dye simply identifies the basal plane of the PM, whereas αSyn signals show pools of endogenous protein at the PM and cell type-specific variations thereof. Brighter patches of PM-dye correspond to preferred incorporation of tetramethylrhodamine-WGA, which does not correlate with (nor represent) PIP2 or αSyn localization. Irrespectively, we selected new image panels to better reflect unrelated WGA and αSyn “staining” in the revised figure (now Figure 1—figure supplement 2C)

Figure 2C: Please justify the use of 100 nm vesicles. If the PM should be mimicked, the vesicles should have a larger diameter.

While we share the reviewer’s concern regarding the (implied) “curvature” aspect of 100 nm vesicles versus larger, giant unilamellar vesicles (GUVs) with a diameter > 1 µm, PIP2-containing GUVs are inherently unstable in the presence of low µM amounts of Ca^2+^ and rupture within minutes^4^, thus precluding experiments presented in Figure 2E. Despite these shortcomings, we believe that 100 nm vesicles (i.e. LUVs) represent excellent surrogates for studying αSyn interactions with “planar” membranes.

Considering the geometries of αSyn and target vesicles as spherical in a first approximation, αSyn molecules approaching these membranes display the following properties:

**Author response image 1. sa2fig1:** 

Given the known diameter of αSyn in solution (~6 nm^5, 6^) and that of our reconstituted vesicles (i.e. ~100 nm), the local interaction surface displays minimal curvature over the entire dimensions of the protein (indicated by arrow heads). Because αSyn residues 1-12 act as the primary anchoring sites^7, 8^, the extension of the corresponding α-helix (~1.6 nm) can be extracted from available experimental data^9^. As can be appreciated in Author response image 1, curvature is negligible over this range of dimensions and initial contacts between αSyn monomers and 100 nm vesicles or planar membranes are virtually identical on the microscopic level. Even in the fully extended helical conformation, spanning roughly 12 nm^10^, curvature effects imposed by 100 nm vesicles are small (not taking membrane bending and remodeling activities by αSyn into account^11^).It is also important to keep in mind that in today’s discussions about αSyn-membrane interactions, “membrane curvature” mostly serves as a proxy for “lipid packing defects”. While it is clear that high curvature induces/exacerbates these defects and enhances αSyn binding, membrane fluidity and composition especially with regard to saturated versus unsaturated fatty acids (FA) act as important additional contributors to packing defects. In this regard, we believe that the poly-unsaturated FA character of most PIPs and induced packing defects exert a stronger influence than curvature itself.

Figure 3A: The quantification of αSyn intensity with expression of INPP5E is not reflected by the provided image where there is essentially no αSyn signal. Please provide a more representative image.

This has been updated and we now provide an image that better reflects residual amounts of endogenous αSyn at the PM upon INPP5E overexpression (new panel in Figure 3A). Specifically, we have chosen a comparative view that shows cells *not* transfected with INPP5E-mCherry (upper left, lower right corners) together with a transfected one in the same frame (center). The differences in PM-localization levels of endogenous αSyn are readily evident (see also Q6 next).

Figure 3A: The authors argue that αSyn has a specificity for PIP2 and PIP3, but why then are there no apparent changes between αSyn intensity upon expression of MTM1 and INPP5E (which result in production of π and PI4P, respectively) versus PTEN (which produces PIP2 from PIP3)? In addition, the authors describe that "only the conversion of PIP2 to PI4P by INPP5E led to a marked reduction" but their statistics show significant changes for all three phosphatases compared to mCherry only.

We fully agree with the reviewer that our *p*-value analysis and corresponding statistical significance (** versus ***) were at odds with our statement. As a matter of fact, they were also at odds with what is shown in the whisker blots (of Figure 3A), as mCherryctrl and MTM1, PTEN datasets were obviously *too similar* to rationalize the low *p*-values. We revisited the source data file and recalculated *p*-values. Indeed, we found that the mCherry-ctrl – MTM1 comparison yields a *p*-value of 0.6297. Therefore, the correct labeling should have been *“not significant”* (NS).

The *p*-value for the mCherry-ctrl – PTEN is 0.0198, thus should have been labeled * (*p* < 0.05). Upon closer inspection, we realized that the significance analysis in this particular case, is dominated by the spread of data points in the control dataset compared to the phosphatase experiments and, therefore, should be considered with caution. By excluding single high-value outliers in the mCherry-ctrl dataset (by adjusting the overall threshold), we learned that the INPP5E experiment displayed the only “robust” value (*** *p* < 0.001).

We rechecked all other *p*-values and now include them in the revised source data files.

In summary, we had mislabeled the significance values in our original submission and updated the figure accordingly. Having made those changes, we believe that our initial statement “In agreement with our hypothesis, only the conversion of PIP_2_ to PI(4)P by INPP5E led to a marked reduction of endogenous αSyn at the PM (Figure 3A).” remains valid. We kept the original wording in the revised versions of the manuscript.

Figure 3: There may be a contribution of coincidental labeling. Could the authors reduce PIP2 and/or PIP3 levels as in panel 3A and repeat their analysis?

Coincidental labeling between PIP2 and PIP3 can be excluded due to the fact that PIP2 detection is antibody-mediated, whereas PIP3 identification is accomplished via transient expression of the GRP1-PH-domain fused to GFP. The affinity of GRP1-PH domain towards PIP3 is 2-3 orders of magnitudes greater than for PIP2, and negligible for other PIPs^12^. Hence, “PIP3-fluorescence” originates from interaction of a PH domain that specifically recognizes PIP3 only. Technically speaking, no cross-reactivity is possible in such a system. We believe that this notion is corroborated by the differential appearance of PIP2 and PIP3 signals in the time course. At 120 seconds, for instance, endogenous αSyn localization “follows” the PIP3 signal, whereas PIP2 fluorescence is virtually absent (Figure 3B). In that sense, we are confident that our experimental setup cleanly reports on signaling mediated redistributions of cellular αSyn in a manner that is PIP2 and PIP3-dependent.

Are the observed CD spectra and NMR spectra shown in Figure 2 consistent with residues ~ 1-100 in an helical conformation and the remaining ~ 40 residues being unstructured under high PIP2 to αSyn ratios, similar to the solution NMR structure of micelle-bound αSyn (PDB ID 1XQ8) or similar to the solid state NMR studies of αSyn interactions with lipid bilayers by Fuso et al., Nature Comm, 2014? In other words, are residues 100-140 unstructured in all these studies? Please comment.

Yes, based on available NMR spectra (Figure 2B and Figure 2—figure supplement 1A and 2B), it is evident that resonances corresponding to residues 100-140, do not undergo line-broadening and do not display chemical shift changes, i.e., superimpose perfectly with those of disordered monomeric αSyn. This corroborates that C-terminal residues remain dynamic and disordered upon PIP2-vesicle binding, similar to all previously determined αSyn-membrane interactions. To strengthen this notion, we modified the text at several points:

Results: “Together, these results established that residues 1-100 of αSyn interacted with PIP_2_vesicles in helical conformations that imposed membrane remodeling, whereas C-terminal residues 100-140 did not engage in membrane-binding and remained flexible and disordered.”

Discussion: “… these interactions are indistinguishable from other previously identified, negatively charged membrane systems with primary contacts by N-terminal αSyn residues 1-10 and progressively weaker interactions along residues 10-100. Cterminal residues 100-140 are not involved in PIP_2_ membrane binding, similar to all other reconstituted vesicular or planar lipid surface interactions studied thus far.”

The experiments with calcium (Figure 2E) suggest that the first ten residues always interact with PC-PIP2 vesicles, although overall binding was greatly reduced. However, plasma membranes contain anionic lipids that may mitigate the effect of calcium that is observed in these experiments. Please comment.

Yes, this is exactly what we intended to recapitulate in this experiment. As outlined in Materials and methods, we used 60 µM of isotope-labeled (^15^N) αSyn and a 50fold *mass* excess of PC-PIP2 vesicles. In molar terms, this corresponds to ~41.25 mM of total lipids and ~2.95 mM of PIP2 given the molar ratio between DOPC:PIP2 of 13:1. We added Ca^2+^ to a final concentration of 2.5 mM. However, membrane-embedded PIP2 can sequester up to 3 Ca^2+^ ions per lipid (3:1)^13^, which indicates that Ca^2+^ is present at substoichiometric amounts with respect to direct PIP2 binding, but in large excess to αSyn.

When αSyn binds to membranes in its characteristic extended helical conformation, it coordinates between 20-30 lipid molecules, depending on lipid type and clustering^14, 15^. If αSyn were to *only* interact with PIP2 molecules on DOPC:PIP2 vesicles, this would translate into 1.2-1.8 mM of PIP2 (at 60 µM of αSyn) and, accordingly, reduce the concentration of membrane-embedded PIP2 not bound by αSyn to 1.75-1.15 mM. At 3:1 of Ca^2+^:PIP2 binding, all of the added Ca^2+^ could coordinate free PIP2 on DOPC:PIP2 vesicles and *not* displace any of the PIP2-bound αSyn. As our results show, this is clearly not the case, i.e. C-terminal portions of αSyn *are* effectively displaced by Ca^2+^.

This strongly argues for a scenario in which αSyn actively clusters PIP2 on the surface of DOPC:PIP2 vesicles (in line with MD simulations ^16^). These PIP2 clusters likely exhibit higher affinity for Ca^2+^ than the remaining portions of non-clustered PIP2. As a result, Ca^2+^ progressively displaces parts of αSyn that exhibit weaker PIP2 interactions (i.e. residues 10100) whereas regions of higher PIP2 affinity are unaffected and remain vesicle-bound (i.e. residues 1-10). In that sense, PIP2 clustering by αSyn is “replaced” by protein-independent divalent metal coordination after the addition of Ca^2+17^. Thus, our experimental setup effectively recapitulates a competitive binding scenario in which “physiological” amounts of Ca^2+^ (within the range of synaptic transmission, for example) compete for anionic lipid interactions with membrane-bound proteins-, in our case αSyn-, that are present at substoichiometric concentrations. This is in line with expected, general behaviors of PIP2 micro-domains in biology^18^.

We did not dwell on this aspect in the original submission because we consider it is outside the scope of the manuscript. We are grateful for being able to communicate it here so that readers can access it in the published *Author Response*.

References

Encinas M*, et al.* Sequential treatment of SH-SY5Y cells with retinoic acid and brain-derived neurotrophic factor gives rise to fully differentiated, neurotrophic factor-dependent, human neuron-like cells. *J Neurochem* 75, 991-1003 (2000).

Lopes FM*, et al.* Comparison between proliferative and neuron-like SH-SY5Y cells as an in vitro model for Parkinson disease studies. *Brain Res* 1337, 85-94 (2010).

Burre J, Sharma M, Tsetsenis T, Buchman V, Etherton MR, Sudhof TC. Α-synuclein promotes SNARE-complex assembly in vivo and in vitro. *Science* 329, 1663-1667 (2010).

Carvalho K, Ramos L, Roy C, Picart C. Giant unilamellar vesicles containing phosphatidylinositol(4,5)bisphosphate: characterization and functionality. *Biophys J* 95, 4348-4360 (2008).

Paleologou KE*, et al.*Phosphorylation at Ser-129 but not the phosphomimics S129E/D inhibits the fibrillation of α-synuclein. *J Biol Chem* 283, 16895-16905 (2008).

Stephens AD, Nespovitaya N, Zacharopoulou M, Kaminski CF, Phillips JJ, Kaminski Schierle GS. Different Structural Conformers of Monomeric α-Synuclein Identified after Lyophilizing and Freezing. *Anal Chem* 90, 6975-6983 (2018).

Bodner CR, Dobson CM, Bax A. Multiple tight phospholipid-binding modes of α-synuclein revealed by solution NMR spectroscopy. *J Mol Biol* 390, 775-790 (2009).

Lokappa SB, Suk JE, Balasubramanian A, Samanta S, Situ AJ, Ulmer TS. Sequence and membrane determinants of the random coil-helix transition of α-synuclein. *J Mol Biol* 426, 2130-2144 (2014).

Fusco G, De Simone A, Arosio P, Vendruscolo M, Veglia G, Dobson CM. Structural Ensembles of Membrane-bound α-Synuclein Reveal the Molecular Determinants of Synaptic Vesicle Affinity. *Sci Rep* 6, 27125 (2016).

Jao CC, Hegde BG, Chen J, Haworth IS, Langen R. Structure of membrane-bound α-synuclein from site-directed spin labeling and computational refinement. *Proc Natl Acad Sci U S A* 105, 19666-19671 (2008).

West A, Brummel BE, Braun AR, Rhoades E, Sachs JN. Membrane remodeling and mechanics: Experiments and simulations of α-Synuclein. *Biochim Biophys Acta* 1858, 1594-1609 (2016).

Kavran JM*, et al.* Specificity and promiscuity in phosphoinositide binding by pleckstrin homology domains. *J Biol Chem* 273, 30497-30508 (1998).

Bradley RP, Slochower DR, Janmey PA, Radhakrishnan R. Divalent cations bind to phosphoinositides to induce ion and isomer specific propensities for nano-cluster initiation in bilayer membranes. *R Soc Open Sci* 7, 192208 (2020).

Galvagnion C*, et al.* Lipid vesicles trigger α-synuclein aggregation by stimulating primary nucleation. *Nat Chem Biol* 11, 229-234 (2015).

Middleton ER, Rhoades E. Effects of curvature and composition on α-synuclein binding to lipid vesicles. *Biophys J* 99, 2279-2288 (2010).

Gambhir A*, et al.* Electrostatic sequestration of PIP2 on phospholipid membranes by basic/aromatic regions of proteins. *Biophys J* 86, 2188-2207 (2004).

Wang YH, Slochower DR, Janmey PA. Counterion-mediated cluster formation by polyphosphoinositides. *Chem Phys Lipids* 182, 38-51 (2014).

Borges-Araujo L, Fernandes F. Structure and Lateral Organization of Phosphatidylinositol 4,5bisphosphate. *Molecules* 25, (2020).